# Easy and Versatile Technique for the Preparation of Stable and Active Lipase-Based CLEA-like Copolymers by Using Two Homofunctional Cross-Linking Agents: Application to the Preparation of Enantiopure Ibuprofen

**DOI:** 10.3390/ijms241713664

**Published:** 2023-09-04

**Authors:** Oussama Khiari, Nassima Bouzemi, José María Sánchez-Montero, Andrés R. Alcántara

**Affiliations:** 1Eco Compatible Asymmetric Catalysis Laboratory (LCAE), Department of Chemistry, Badji Mokhtar University, Annaba 23000, Algeria; oussama.khiari@gmail.com (O.K.); nbouzimi@yahoo.fr (N.B.); 2Department of Chemistry in Pharmaceutical Sciences, Pharmacy Faculty, Complutense University of Madrid (UCM), Ciudad Universitaria, Plaza de Ramon y Cajal, s/n., 28040 Madrid, Spain

**Keywords:** CLEA-like copolymers, one-pot immobilization, cross-linkers, pH jump, amine activation, ibuprofen resolution, enzymatic hydrolysis

## Abstract

An easy and versatile method was designed and applied successfully to obtain access to lipase-based cross-linked-enzyme aggregate-like copolymers (CLEA-LCs) using one-pot, consecutive cross-linking steps using two types of homobifunctional cross-linkers (glutaraldehyde and putrescine), mediated with amine activation through pH alteration (pH jump) as a key step in the process. Six lipases were utilised in order to assess the effectiveness of the technique, in terms of immobilization yields, hydrolytic activities, thermal stability and application in kinetic resolution. A good retention of catalytic properties was found for all cases, together with an important thermal and storage stability improvement. Particularly, the CLEA-LCs derived from *Candida rugosa* lipase showed an outstanding behaviour in terms of thermostability and capability for catalysing the enantioselective hydrolysis of racemic ibuprofen ethyl ester, furnishing the eutomer (*S*)-ibuprofen with very high conversion and enantioselectivity.

## 1. Introduction

During the last decade, applied biocatalysis has proven to be a key enabling factor for the preparation of drugs and fine chemicals [1,2,3,4,5,6]. The improvement of some of the properties of enzymes can be addressed using molecular biology techniques (metagenomics, directed evolution, protein engineering), which has led to great advances in the design of biocatalysts [7,8,9,10,11]. Comparing all the different strategies used, immobilization is a smart solution to the problem of solubility and reuse, leading to a concomitant decrease in the costs of the process [12,13,14,15,16,17,18,19,20,21,22,23]. On the other hand, a great effort has been dedicated to the design of new immobilization strategies to solve this problem and simultaneously improve other properties. For multimeric enzymes, stability can be improved if a multipoint or multisubunit covalent bonding network is achieved, or if a favourable microenvironment is generated (i.e., causing partitioning of inactivating molecules) [24,25,26,27,28]. Hydrolases are a very important class of enzymes because of their wide versatility and because they catalyse a large number of reactions [29,30,31,32]. Within this category, lipases have demonstrated great versatility, efficiency, selectivity, low cost, biodegradability and interesting industrial applications in different areas such as food, pharmaceutical, cosmetic, textile and biotechnology [33,34,35,36,37,38,39,40,41]. Thus, hydrolases (mainly lipases) are the most frequently immobilised enzymes [29,30,42,43,44,45,46].

Different materials can be used to carry out the immobilization of enzymes [45,47]. In this sense, the use of organic compounds from natural sources to produce stable solids with a suitable surface can be carried out using polymers or as complexes through the union of particles with a chemical reaction [48,49]. In the literature, some examples, such as the production of stationary chiral phases or the immobilization of enzymes, can be found [50]. Of all the existing methodologies in the literature, with the wide collection of supports used (inorganic and organic types) and using all existing types of activation of the supports to carry out an effective binding between support and enzyme, very few of them have shown a high effectiveness in relation to increasing the activity of the biocatalyst [46].

Glutaraldehyde (GA) is a homobifunctional reagent commonly used for immobilizing enzymes mainly through lysine’s terminal amino group [51]; this can be done either by linking the enzyme to amine-containing solid supports [52,53,54,55] or using GA as an amine-reactive homobifunctional cross-linking agent [55,56]. This last technique is the base of the preparation of CLEAs (Cross-Linked Enzyme Aggregates), a type of enzyme preparation in which the enzyme molecules are precipitated and covalently linked together to form a solid, insoluble material [12,42,57]. This can be performed either through a chemical cross-linking process, such as the one promoted by GA, or by physically aggregating the enzymes through several methods such as spray drying [12]. CLEAs have several potential benefits over traditional enzyme preparations, including increased stability, longer shelf life, and the ability to be used in continuous flow reactions [12,57,58]. They may also have improved catalytic activity and substrate specificity compared to the individual enzymes. However, the preparation of CLEAs can be challenging and may require specialised techniques [59,60,61].

In fact, the reaction with glutaraldehyde provides binding arms with a carbonyl end capable of reacting mainly with amino groups (although reactions with other enzymatic functional groups such as thiol, phenol or imidazole are also possible). The exact nature of the chemical bonding between an enzyme’s active residues and GA is not clear, because of the different chemical structures that GA can adopt in aqueous solutions, which would lead to different chemical bonds [51,55]. In any case, the simplest chemical reaction between GA and lysine residues would imply the formation of labile imine groups, therefore requiring a soft chemical reduction (generally using NaBH_4_ or NaBCNH_3_) to furnish stable amine bonds.

To produce an effective immobilization of enzymes through lysine amino groups, a high amount and correct orientation/exposition of these residues is crucial. If the lipase does not fulfil any of these requirements, several strategies must be employed to solve this problem; thus, genetic modification or chemical amination, generally by using ethylenediamine or carbodiimide [62], can be used. When intending to perform the immobilization using CLEAs, it is frequent to add a co-feeder, e.g., a lysine-rich inert protein (such as BSA) or a polyamine, just to increase the number of reactive spots and to promote the cross-linking.

As an example of the last method, in this paper we describe an easy preparation of cross-linked enzyme aggregate-like copolymers (CLEA-LCs) using a combination of glutaraldehyde and putrescine as cross-linkers. Putrescine (1,4-butanediamine), biosynthesised in vivo from arginine, which can easily react with aldehyde groups such as those present in glutaraldehyde, was used to form covalent bonds and help promote an enzymatic cross-linking. Thus, six different lipases were used to generate the corresponding CLEA-LCs using one-pot, consecutive cross-linking steps using glutaraldehyde and putrescine through means of a pH alteration (pH jump) to promote amine activation. The general scheme is summarised in Figure 1. The thermal stability and reusability of the best CLEA-LCs, based on lipase from *Candida rugosa*, were assessed. To check the practical applicability of the CLEA-LCs, their capability to catalyse the resolution of racemic ibuprofen ethyl ester to furnish enantiopure (*S*)-ibuprofen (the eutomer of an essential member of non-steroidal anti-inflammatory drugs (NSAIDs) used for the symptomatic treatment of various types of arthritis (i.e., rheumatoid arthritis, osteoarthritis and ankylosing spondylitis) [63,64]) was tested.

## 2. Results

### 2.1. Preliminary Adjustment of Enzyme and Cross-Linkers Amounts for Further CLEA-LC Preparation

As a first preliminary investigation of the described method, the effect of the amount of glutaraldehyde (GA) on the specific activity of CAL-B was assessed. For this purpose, different amounts of GA (ranging from 1 to 20 V%) were added to 10 mL of 0.5 mg/mL of CAL-B and stirred for one hour at room temperature, measuring the specific activity after this incubation (see Section 4.2.4). The results are shown in Figure 2.

In a subsequent step, a study of the best combination of enzyme, glutaraldehyde and putrescine (as PTR-HCl pH 6.5) was carried out in order to establish the best proportions of the three reagents, as reported in Section 4.2.4). The results are shown in Table 1.

As can be seen from Table 1, conditions G4–10 (bold, italics) provided the best results, and they were used for the further experiments.

### 2.2. In-Depth Study of CLEA-like Copolymerization Procedure as a Versatile Technique

To better understand this technique, the previous procedure was also applied to five other lipases, apart from CAL-B: CRL (lipase from *Candida rugosa*), CAL-A (NovoCor^®^AD L, lipase A from *Candida antarctica*), TTL (lipase from *Thermomyces lanuginosus*), RML (Palatase 20000, lipase from *Rhizomucor miehei*) and LU (chimera produced through the fusion of the genes of the lipase from *Thermomyces lanuginosus* and the phospholipase A1 from *Fusarium oxysporum*). As stated before, the best conditions obtained for CAL-B (Table 1, G4 case 10, molar equivalents 1:2000:1000 (CAL-B.:GA:PTR) were used, as reported in Section 4.2.5. GA-activated enzyme (form S1) monomers and dimers in general (possible polymer existence) are shown in SDS-page electrophoresis shown in Figure 3 as reported in Section 4.2.6

As presented in Section 4.2.5, at the end of step 1, PTR-HCl (pH 6.5) was added to the reaction mixture. The addition of this partially inactive hydrochloride form was selected in order to avoid three main problems: (i) the characteristic undesirable odour of putrescine; (ii) its alkaline character could damage/denature enzyme molecules if directly added as PTR; and (iii) to avoid any possible fast and localised reaction (imine formation) directly with glutaraldehyde due to its high nucleophilicity. In this way, PRT-HCl molecules could be distributed homogeneously and slowly react during step 2 of the process. The changes in the specific activity of derivatives are shown in Figure 4. Subsequently, pH adjustment (pH jump step) was performed using the dropwise addition of 1 M NaOH aqueous solution. Figure 5 shows the detailed values for CRL CLEA-LC.

Detailed data regarding the pH jumps for all preparations are displayed in Table 2. As a special case, CAL-B* was the result of a second pH jump from 8.19 to 12.0. Most of the pH jumps were carried out from pH values between approximately 6.0 and 8.0. It was found that this step had no notable effect on enzyme structure, as deduced from the increase in specific activity at the end of step 3 in almost all of the studied cases.

Table 3 shows the characterization of the different derivatives obtained after pH jumps for each enzyme (passage from step 2 to step 3, while in Figure 6 the results of the corresponding residual specific activities (R.S.A) and recovered (R.A) activities are shown.

At the end of step 3, the formed solid catalyst (S3) was recovered through centrifugation (8000 rpm, 15 min) and washed using Milli-Q ddH_2_O, to furnish the catalyst as a dark-brown compacted precipitate with a high percentage of retained water (approximately 93%, Table 3). Figure 7 shows the final forms of the catalysts for CRL. S3 was the catalyst at the end of the third step, as a yellowish coloured precipitate which was recovered and compressed through centrifugation (8000 rpm, 15 min) to obtain a dark-brown solid with high content of entrapped water (up to 93%). Form (or state) S4 of the final catalyst was obtained through treatment with 1 mg/mL NaBH_4_, while no (or a negligible) effect of this procedure on specific activity of enzyme was observed.

### 2.3. Characterization of CLEA-LC Using FTIR Spectroscopy and SEM

FTIR spectroscopy was performed to detect the formation of imine function that comes from the reaction of glutaraldehyde with amines (aldehyde moieties on enzyme surface reacting with PTR). The characteristic absorbance band of imine –N=C– was observed at 1637 cm^−1^ (spectre of CRL CLEA-LC, Figure 8A) and a notable decrease in the band of –N–H on the spectre of free CRL (1067 cm^−1^, Figure 8B). These results indicate the formation of Schiff bases between aldehyde and amine functions of the three components of the catalyst.

The SEM images shown in Figure 9 for CLEA-LC from CRL reported an amorphous surface for the enzymatic preparations, with a particle size of approximately 100–200 µm with the presence of randomly distributed pores.

### 2.4. Thermal Stability of CRL CLEA-LC

One of the most attractive findings using the described technique was the observed thermal stability of the CLEA-LC derived from CRL. Thus, the deactivation plots of the CLEA-LC were studied and compared to the free enzyme at different temperatures (25 °C, 40 °C, 50 °C, 60 °C and 70 °C). The results are shown in Figure 10.

### 2.5. Long Term Storage Stability at 4 °C

As described in Section 4.2.13, CRL and CAL-B CLEA-LC (500 mg from S4 of each CLEA-LC) were studied to test their storage stability at 4 °C for 3 months (90 days) and compared to the free enzymes by analysing their hydrolytic activities versus *p*-NPB (see Section 4.2.3). The results are shown in Figure 11.

### 2.6. Application of CLEA-LCs in the Kinetic Resolution of Rac-Ibuprofen Ethyl ester

The CLEA-LCs of the six studied enzymes, as well their native counterparts, were measured for their capability to perform the enantioselective hydrolysis of *rac*-ibuprofen ethyl ester during 24 h at 37 °C (Figure 12). The results are shown in Table 4.

### 2.7. Reuse of CRL CLEA-LC for the Kinetic Resolution of Rac-Ibuprofen Ethyl Ester

The CLEA-LC of *Candida rugosa* lipase was subjected to five cycles of reuse in the same conditions as mentioned in Section 2.6 to test its ability to retain its catalytic properties. The results are shown in Figure 13. A small diminution of enantiomeric excess values (cycle 1 and 2: 99%, cycle 3: 95%, cycle 4:94%, cycle 5 93%), alongside a moderate decrease in conversion from the third cycle indicated that the obtained catalyst could maintain the conformation required to catalyse the hydrolysis of *rac*-ibuprofen ethyl ester with a small loss in catalytic units.

## 3. Discussion

As commented in the introduction, the most used carrier-free immobilization method is the preparation of CLEAs; a plethora of publications and reviews have described CLEAs with multiple enzymes [12,42,57,58,67]. This methodology involves the precipitation of the enzyme molecules (with or without other reagents) and the ulterior crosslinking using mainly (but sometimes not only) GA [51]. Nevertheless, other carrier-free methods not requiring the initial precipitation step, such as the generation of copolymers, have rarely been studied [68]. Therefore, our aim was to study the possibility of preparing lipase-based cross-linked enzyme aggregate-like copolymers (CLEA-LCs) by simply interconnecting lipase molecules with one (GA) or two (GA/PTR) homobifunctional linkers and checking whether this simple and fast methodology could produce active and stable biocatalysts.

### 3.1. Preparation of CLEA-LCs

Lipase B from *Candida antarctica* (CAL-B) was used as a model enzyme during the preliminary investigations because of its low amount of superficial lysine residues (Lys 136, Lys271, Lys290 and Lys308 in the active site/lid face; Lys 13, Lys32, Lys98 and Lys124 the opposite face [69,70]), which impedes the traditional cross-linking procedure using glutaraldehyde and demands, in most cases, the addition of other amine-containing macromolecules or proteins as co-feeders for CLEA preparation [51,71,72]. The results shown in Figure 2 indicated a positive correlation between the amount of GA and the specific activity at a low GA percentage of up to 5% V, reaching almost a two-fold increase in the specific activity; for higher GA percentages, a relative decrease in activity was observed, leading to a plateau showing activated derivatives (approximately 45–50% increase) compared to the unmodified enzyme. Thus, the slight chemical modification of Lys groups in the CAL-B structure seemed to be positive for increasing the specific activity of this enzyme, while higher amounts of GA could lead to any kind of intra- or inter-crosslinking. In fact, it is well known that for conventional homobifunctional cross-linkers, the ideal scenario is the modification of half of the reactive groups to promote cross-linking, but for GA, as amino/GA moieties reacts better with other amino/GA groups, the use of moderate GA concentrations (even suboptimal) is the appropriate choice [51].

During a second evaluation study, putrescine (butane-1,4-diamine) was introduced as a second cross-linker agent which could attach the GA-activated enzyme by means of imine bonds. As far as we know, this is the first time that this methodology has been reported in the literature; in fact, even though the use of GA and PTR for immobilizing enzymes had been reported for attaching enzymes to membranes [73,74], no previous studies on their combined use for preparing CLEAs could be found in literature. According to the obtained results presented in Table 1, the combined use of the reagents (GA and PTR) did not affect the specific activity of CAL-B (initial value 35 U/mg, very similar in all cases considering the experimental error, which was approximately 5%). The activity yield (A.Y, %) was utilised as a comparative parameter to assess the impact of the immobilization technique on the enzyme catalytic properties, by comparing the activities of the same enzyme before and after being incorporated into the CLEA-LC core. In almost all cases we observed an excellent retention of enzyme catalytic properties attending to this parameter, as shown in Table 1. Regarding immobilization yield (I.Y, %), the best results (also for A.Y.) were obtained using the combination G4-10, which was therefore selected for the subsequent in-depth study of CLEA-LCs for other lipases. Indeed, a slight decrease in the activity yield of A.Y and I.Y. was observed in the case G4-11, maybe due to an over multipoint attachment of enzyme particles which could affect the enzyme structure and conformation. On the other hand, using an excess of reagents may lead to over-copolymerization between GA and PTR without including enzyme particles as observed in I.Y for the case G4-11 (Table 1). Therefore, and according to these considerations, the experimental conditions for combination G4-10 (molar equivalents 1:2000:1000 (lipase:GA:PTR) were assumed to be optimal and were employed in the preparation of CLEA-LCs with other lipases.

As mentioned above, the best conditions obtained for the preparation of CLEA-LCs for CAL-B were applied to the other lipases studied. The first step was the treatment with GA. During this step, free enzymes were modified using GA in order to create enzyme monomers, dimer and polymers where possible. The 12% SDS page electrophoresis (Figure 3) showed that the dominant forms were GA-activated monomers and dimers (enzyme units with aldehyde functions on the surface) after 1 h of treatment. According to the results shown in Figure 4, a notable increase in specific activity was observed in the case of CAL-A (156%), while this increase was less pronounced for CAL-B (124%); nevertheless, a notable decrease in specific activity was observed for the S1 forms of TLL (61%), LU (69%), RML (50%) and CRL (53%) at the end of step 1. This could have been caused by the fact that these enzymes were more affected by the multipoint attachment due to the presence of a higher number of lysine residues on their surfaces [75]. In any case, the initial activity of CRL was an order of magnitude higher than those obtained with the other enzymes, as they had a different origin and physic state (see Section 4.1.1).

In the second step (PTR·HCl addition, creation of S2 state), the same trend was found for CRL, with an important decrease in specific activity (22% of the free CRL) observed at the end of step 2 (Figure 4), which could be explained by the over-cross-linking in the presence of both GA and PTR. CAL-A, TLL and LU also led to less active derivatives, while RML and CAL-B showed an increased activity compared to their S1 forms. Finally, the pH jump led in almost all cases to derivatives (S3) possessing a similar specific activity compared to S0, except for CRL, which showed a slightly lower activity, retaining approximately 75% of its initial activity. In the case of CAL-B* (CAL-B CLEA-LC after a second pH jump from 8.19 to 12.0, Table 2) we observed a notable increase in the immobilization yield (from 35 to 99%) and specific activity (from 35 to 40 U/mg) with an observed high pH stability.

Table 3 covers the results obtained at the end of the CLEA-LCs preparation (form S3); we obtained the highest immobilization yields (I.Y., %) for CRL and RML, with more than 99%, which was possibly due to the higher content in the lysine residues on their surfaces. Considerably high yields (from 72 to 83%) were observed for CAL-A, TLL and LU, while the lowest yield was observed in the case of CAL-B (35%), probably due to the well-established low amount of lysine residues on its surface. This problem was solved by making an additional second pH jump (8.19–12.0) to obtain CAL-B* with a higher yield of more than 99%.

The described technique has shown no (or little) effect on the catalytic properties of the utilised enzymes traduced by comparing specific activities of the free form and the CLEA-like copolymer (S3* form: compressed S3, Table 3). This point was shown in Figure 6 by quantifying the activity yield (A.Y., %) as a comparative parameter to measure the activity of the incorporated enzyme particles before and after being immobilised. A good retention of activity was observed in cases CAL-B, CAL-B*, TLL, LU and RML with an A.Y of 106%, 114%, 106%, 100% and 107%, respectively (Figure 6). A small decrease was observed in the cases of CRL (76%) and CAL-A (77%) which could be attributed to the effect of reagents on the structure in general and the diffusion limitation promoted by the impact of cross-linkers on the active sites of these two enzymes.

The recovered activity (R.A. %, also shown in Figure 6) was also an important comparative parameter to be considered, in order to show how the procedure was efficient by considering the total units in the starting suspension (catalytic units present in 50 mg of initial enzyme) and the total units observed in the CLEA-LCs at the end of immobilization procedure. The highest R.A (%) was observed in the cases of CAL-B* and RML (113.14% and 106%, respectively). The rest of the enzymes, except CAL-B, showed good to high recovered activities from 64 to 76%. The lowest recovered activity was observed in the case of CAL-B (37%) which can be explained as it was previously (low lysine residues on the surface).

On the other hand, the trapped water percentage (Table 3) was approximately 93% for all CLEA-LCs after compression through centrifugation (8000 rpm, 15 min), as a main part of the catalyst was found to be crucial for the functionality and resistance of the CLEA-LC. Interestingly, the total removal of this water by leaving the catalyst in the open-air for 24 h led to a total loss of catalyst functionality. Thus, this core water content as part of the catalyst enabled a good resistance and shelf life as long as it remained in the set.

Figure 7 shows the final aspects of CLEA-like copolymers (CRL CLEA-LCs were taken as an example). The pale-yellowish colour of the catalyst at the end of the immobilisation process (S3) turned to a dark-brown colour after compression through centrifugation (S3*), which may have been due to the concentration of imine functions which absorbed more light than the suspended one. After treatment with NaBH_4_, the catalyst recovered its yellowish colour (S4, Figure 7) after the reduction in imine functions

Therefore, from all the data presented so far, we decided to focus our subsequent studies mainly on the CLEA-LC based on *C. rugosa* lipase, as it showed the most interesting properties. We must underline that CRL was not the enzyme most generally employed for producing CLEAs (see recent review from Sampaio et al. [42]), and never before for the preparation of polymeric carrier-free derivatives.

### 3.2. Thermal Stability of CRL CLEA-LCs

The thermal stability under operational conditions was measured for CRL CLEA-LCs and their free counterparts by monitoring the residual activities in *p*-NPB hydrolysis (%) over 384 h (16 days). The experimental results at the different temperatures tested (25, 40, 50, 60 and 70 °C) are shown in Figure 10. The experimental data shown in Figure 10 were fitted to a sum of exponential decay functions (Y = An·(exp(−kn)·X), using the software EXFIT implemented inside the SIMFIT fitting package (version 7.6, Release 9), a free-of-charge open-source software for simulation, curve fitting, statistics and plotting (accessible at https://simfit.uk/download.html, accessed on 31 August 2023)). Table 5 shows the results obtained from the fitting.

As can be seen from Figure 10 and Table 5, a significant improvement was observed in the thermal deactivation for the CLEA-LCs compared to the native enzyme. In fact, there was no noticeable decrease in activity at 25 °C in the case of CLEA-LC during the period of study compared with a static half-life of 192 h (8 days) in the case of the free CRL. The static half-lives at 40 °C, 50 °C and 60 °C were 312 h, 96 h and 2 h in the case of CLEA-LC compared to 35 h, 0.67 h (40 min) and 30 min in the case of free CRL. At 70 °C, a quick deactivation was observed in the case of CLEA-LC (total loss of activity after approximately 1 h), but again the behaviour was better than that of the free enzyme, which had no activity whatsoever.

The thermal stability observed with CRL CLEA-LC indicated that the covalent multipoint attachment obtained more stable enzyme molecules compared with its free counterpart. This gave it the possibility to be applied in the reaction which require more heating up to 60 °C with an observed shelf life of 5 days compared to 40 min for the free CRL at this temperature (Figure 8). Despite the low resistance at 70 °C with a shelf life of 3 h compared to an immediate loss of activity for the free CRL, the CLEA-LC from CRL is not recommended for use at this temperature.

Considering not only the half-life time but also the deactivation profiles, the behaviour of CLEA-LCs was different compared to the free CRL. The fitting to a series of exponential decay functions can be used to report different deactivation steps, according to the archetypal series-type mechanisms proposed by Henley and Sadana [76,77], involving multiple steps leading to the loss of enzymatic activity. Thus, different deactivation models were proposed and classified into two major categories, depending on the possibility of having or lacking a residual activity greater than the initial activity at any time (not necessarily near t = 0); in all cases, both for CLEA-LCs and native CRL, an increase in residual activity compared to the initial activity was never observed, so in that case we found ‘case a’) according to the Henley–Sadana categorisation [76]. For the free CRL, interestingly, the behaviour at 25 °C could be typified as case 6, as there was a grace period (48 h) in which the enzyme apparently suffered no loss or deactivation; after that period, the deactivation followed a complex decay which could be simulated using a three-step model, the constants of which are shown in Table 5. At 40 °C, a much shorter grace period (7 h) was observed, followed by a one-step single exponential decay. For 50 °C and 60 °C, no grace periods were detected, and gradually faster single decays were reported, while at 70 °C the deactivation was too fast to be detected.

As commented before considering the half-life times, CLEA-LCs were much more stable toward thermal deactivation. In fact, at 25 °C no noticeable deactivation was observed after 16 days, while at 40 °C, the deactivation could be fitted to a three-step model, with a sort of concave behaviour (case 5 according to the Henley–Sadana model [76]), characterised by an initial k_1_, which is one order of magnitude higher than k_2_ and k_3_, which are similar. At 50 °C, a similar pattern was observed, but the deactivation was faster as the three kinetic constants were similar. Finally, at 60 and 70 °C, fitting to double exponential decay was observed, although at 70 °C the deactivation was very fast.

Taking into account the long-term storage stability at 4° depicted in Figure 11, free preparations of CAL-B and CRL showed a half-life time of 15 and 18 days, respectively, under the storage conditions (Figure 11), with grace periods of 4 and 6 days, respectively, and similar deactivation profiles (single exponential). Meanwhile, both CLEAs showed only a slight decrease in their specific activities (approximately 8%) after three months, with grace periods of one month for the CAL-B CLEA-LC and 2 months for the CRL counterpart. Thus, this storage stability clearly demonstrated the efficiency of this technique to furnish stable catalysts able to maintain their functional integrity for a good period. Additionally, the short time required for their preparation (around 2 h) undoubtedly make this an attractive pathway to obtain immobilised catalysts.

Compared with the other latest examples of CLEAs with CRL found in the literature, our CLEA-LC seems to be advantageous; for instance, Jiaojiao et al. [67] recently reported CRL-CLEAs using an expensive amine-containing ionic liquid (1-aminopropyl-3-methylimidazole bromide, FIL) as the functional surface molecule with which to modify CRL. The activity of this CLEA was tested only with a chromogenic standard substrate without testing it on any racemic substrate to check its stereoselection capability. Furthermore, the thermal stability (tested only at 35 °C) was very poor, losing 50% of the activity after only 50 min, and its reusability was only moderate, with approximately 40% residual activity after five cycles.

### 3.3. Application and Reuse of CLEA-LC

As commented in Section 2.6, CLEA-LCs of the six studied enzymes, as well their native counterparts, were tested for the enantioselective hydrolysis of *rac*-ibuprofen ethyl ester during 24 h at 37 °C. According to the results shown in Table 4, the stereobias of the lipases was different: actually, five of them (CRL, CAL-A, RML, LU and TLL) showed (*S*)-stereopreference in the hydrolysis of the racemic ethyl ester, thus leading to the desired (*S*)-eutomer. In contrast, CAL-B and LU led to the (*R*)-distomer. This stereobias for the hydrolysis of racemic esters from ibuprofen is well-documented in the literature [78,79,80,81,82], and specifically for the different lipases CRL [83,84,85,86,87], RML [79,88,89], CAL-B [90,91], CAL-A [92] and TLL [81,93], but not for LU.

Free CRL showed an outstanding performance, leading to an excellent kinetic resolution, showing almost a total conversion (49% vs. 50%, the theoretical maximum for a kinetic resolution) with an E value higher than 200. This enantiomeric ratio E, although commonly used and accepted, is not easy to directly interpret, as it involves logarithmic scales (see Table 4 footnote). As reported, E values below 15 are unacceptable for practical purposes, with moderate to good values in the range of 15–30, and anything above this value considered excellent [94]). Thus, E > 200 indicates that free CRL was acting with exquisite (*S*)-stereoselectivity on the hydrolysis of the racemic ester. Regrettably, the rest of the free enzymes did not show an adequate resolution capability.

When using the CLEA-LCs, a clear improvement was observed in almost all cases (except CAL-B) compared to their free counterparts in terms of enantiomeric excess (%) and/or conversion (%). Hence, a significant improvement was observed in the case of CAL-B* (conversion growing from 25 to 45%, with ee_p_ growing from 15 to 48%), LU (conversion from 8 to 33% and enantiomeric excess of the product (*R*)-ibuprofen from 6 to 47%) and CAL-A; for this latter one, an increase in conversion from 23 to 46% was observed, with an inversion of the enantiopreference from (*S*) to (*R*), along with an observed increase in the enantioselectivity from 3 to 14.3 (moderate-to-good enantioselectivity). The inversion of the enzymatic stereobias upon immobilisation in CAL-A has been observed in some cases because of the peculiar structure of this enzyme [95]. As for the free enzymes, the CLEA-LC from CRL proved to be the best catalyst, maintaining a similar conversion (40% vs. 49%) and even increasing in enantioselectivity (from 97 to 99% for ee_p_). Similar results in terms of increasing the enantioselectivity of CRL on other substrates by using enzyme-aggregates were previously reported [96]. In any case, the outstanding enantioselectivity obtained with the CRL CLEA-LC in the enzymatic hydrolysis of *rac*-ibuprofen ester was higher than those reported for other immobilised CRL-derivatives, such as covalently immobilised CRL on epichlorohydrin-coated magnetite nanoparticles [85], adsorption on macroporous adsorptive resins [97] or CRL covalently immobilised on epoxy-functionalised silica [79].

Recently, Salgin et al. [98] described the use of magnetic and non-magnetic cross-linked aggregates based on CRL for the kinetic resolution of a similar substrate (methyl ester of racemic naproxen (2-(6-methoxynaphthalen-2-yl)propanoic acid) via hydrolysis in a biphasic medium buffer/isooctane. A high enantiomeric excess (close to 100%) was obtained after 40 h, although no conversion data were reported, so that a direct correlation with the behaviour of our CRL CLEA-LC could not be established.

The reusability of the CRL CLEs-LC is one of the most important goals of the immobilisation besides its thermal stability. As shown in Figure 13, CRL CLEA-LC showed a notable retention of catalytic properties for up to five cycles in terms of conversions and remarkably for enantiomeric excess, which only decreased from 99% to 93% after five reaction cycles. Remarkably, even though the conversion dropped from 40 to 22% after the fifth reaction cycle, the excellent thermostability shown by the CRL CLEA-LC (data shown in Figure 10 and Table 5) would enable the maintenance of the productivity by simply duplicating the reaction time.

As stated before, other CRL-CLEAs exhibited poorer behaviour, displaying low reusability of approximately 40% residual activity after five cycles [67]. On the other hand, Sampath et al. [99] recently reported the preparation of a CLEA created using GA to cross-link CRL molecules, using bovine serum albumin (BSA) and polyethileneamine (PEI) as the co-feeders. This catalyst was tested in the hydrolysis of triglycerides containing n-3 polyunsaturated fatty acid moieties, showing an improved catalytic behaviour compared to free CRL, with a moderate reusability up to seven reaction cycles. However, compared to our CLEA-LC, their methodology is more expensive, as it requires two co-feeders (BSA and PEI) instead of the one (PTR) used for our copolymers. For the CLEAs reported by Salgin et al. [98], the authors reported that 93% and 81% of the original value of ees% remained after four cycles for M-CLEAs and CLEAs, respectively. Thus, as can be seen in Figure 13, our CRL-based CLEA-LC showed a better behaviour, as the retention of ee% after five cycles was 93%.

## 4. Materials and Methods

### 4.1. Materials

#### 4.1.1. Enzymes

Lipase from *Candida rugosa* Type VII, ≥700 U/mg (CRL) was purchased from Merck Life Science S.L.U. (Madrid, Spain). Lipase B from *Candida antarctica* (Lipozyme^®^, CAL-B), lipase A from *Candida antarctica* (NovoCor^®^AD L, CAL-A), lipase from *Rhizomucor miehei* (Palatase 20000; RML), lipase from *Thermomyces lanuginosus* (TLL) and Lecitase^®^ultra (LU, chimera produced through the fusion of the genes of the lipase from *Thermomyces lanuginosus* and the phospholipase A1 from *Fusarium oxysporum*) were kindly gifted from Novozymes Spain.

#### 4.1.2. Chemicals

Both 1,4-diaminobutane (putrescine, PTR) and *rac*-ibuprofen were purchased from TCI Europe (Paris, France). Glutaraldehyde (GA) grade II (25% in H_2_O) was purchased from Merck Life Science S.L.U. (Madrid, Spain). Bradford reagent from Bio-Rad (Alcobendas, Madrid, Spain). HPLC solvents were purchased from Scharlab S.L. (Colmenar Viejo, Madrid, Spain).

### 4.2. Methods

#### 4.2.1. Protein Quantification

Protein was quantified using Bradford’s method [100]. In a 1.5 mL Eppendorf tube, 200 µL of commercial Bradford’s reagent were mixed with 800 µL of ddH_2_O-Milli-Q water using a vortex stirrer. Then, 20 µL of enzyme suspension with pre-adjusted concentration under the interval from 1 to 0 mg/mL was added. The suspension was vortexed for a few seconds and the absorption at 595 nm was read after 5 min. A calibration curve of BSA (0 to 1 mg/mL) with R^2^ > 0.99 was used to determine enzyme concentration.

#### 4.2.2. Preparation of Putrescine HCl pH 6.5 Aqueous Suspension (PTR-HCl)

A total of 3 g of putrescine (previously melted at T > 27.5 °C) was added to 10 mL of ddH_2_O Milli-Q; then, 1 M and 1.5 M HCl aqueous solutions were used to adjust the suspension to a final volume of 100 mL and final pH 6.5, in order to obtain a 30 mg/mL (0.34 mmol/L) final concentration of putrescine.

#### 4.2.3. Enzymatic Activity Assays

Hydrolytic activity of the enzyme samples obtained in the different steps of CLEA-LC preparations was tested using *p*-nitrophenyl butyrate (*p*-NPB) as substrate using an internal standard method [1]. In brief, 50 µL of a solution of *p*-NPB (1.2 mM in dry acetonitrile) was added to 2.5 mL of 25 Mm sodium phosphate buffer pH 7.0; then, 50 µL of enzyme preparation with an adequate dilution for enzyme assay was added. The progress curve following the release of *p*-nitrophenol (*p*-NP) was followed for 90 s at 25 °C and 348 nm (isosbestic point of *p*-NP [101]) with internal steering. The molar extinction coefficient ɛ = 5150 M^−1^ cm^−1^ was previously obtained using *p*-NP calibration curve. All experiments were carried out in triplicate.

#### 4.2.4. Preliminary Preparation of CLEA-LC Using CAL-B As Model Enzyme

CAL-B (≈33 kDa) was used as a model enzyme for preliminary adjustment of reagents in a screening step, as most of the investigated enzymes have a close molecular weight (≈30 kDa), except CRL, which has a molecular weight approximately double that of CAL-B. Firstly, the effect of glutaraldehyde on CAL-B was studied by adding an increasing volume percentage (V%), from 1 V% to 20 V% of glutaraldehyde to 10 mL of a suspension (0.5 mg/mL) of CAL-B in 25 mM sodium phosphate buffer pH 7.0. The mixture was stirred for 1 h at 120 rpm and specific activities were determined, as mentioned in Section 4.2.3.

Subsequently, different amounts of GA and PTR (ranged from 250 to 2000 equivalents from each reagent) were added to 11 suspensions of 10 mL of 0.5 mg/mL of CAL-B (5 mg of CAL-B protein, Bradford, 0.15 µmol), as schematised in Table 1. For evaluating the efficiency of the process, three immobilization parameters were calculated: immobilization yield (I.Y (%)), specific activity of CLEA-LC (S.A CLEA-LC) and activity yield (A.Y. (%)) by using the following formulae already reported [102].
$$\mathrm{Immobilization}\;\mathrm{yield}\;(\%):\;I.Y(\%)=\frac{\left(A0-At\right)}{A0}\times 100$$

A0: initial volumetric activity of the suspension (U/mL).

At: volumetric activity in the suspension after the CLEA-LC formation (U/mL).
$$\mathrm{Specific}\;\mathrm{activity}(U/mg):\;S.A(U/mg)=\frac{\mathrm{observed}\mathrm{activity}\;(µ\mathrm{mol}/\min)}{\mathrm{amount}\mathrm{of}\mathrm{used}\mathrm{enzyme}\;(\mathrm{mg})}$$
$$\mathrm{Activity}\;\mathrm{yield}(\%)\;A.Y(\%)=\frac{total\;units\;in\;the\;CLEA-LC}{total\;units\;of\;free\;CAL-B\;incorporated\;in\;the\;CLEA-LC}\times 100$$

Molar equivalence (CAL-B:GA:PTR) was calculated by considering the following data: the molecular weight of CAL-B (33 kDa), equivalence between weight in mg and Da, Avogadro number, density of GA and water and molecular mass of GA and PTR.

#### 4.2.5. Preparation of CLEA-LC Using Six Enzymes

All the preparations mentioned in this study (except CAL-B*, obtained after a second pH jump from 8.10 to 12.0) were prepared using the combination G4-10 (Table 1, case 10, molar equivalents 1:2000:1000 (CAL-B.:GA:PTR)), as they led to best results using CAL-B in the previous screening (Section 4.2.4). Thus, 50 mg from each of the six enzymes used in this study (CRL, CAL-B, CAL-A, TLL, RML and LU) were suspended in 25 mM sodium phosphate buffer pH 7.0 at a final volume of 100 mL and final concentration of 0.5 mg/mL. Then, 1.2 mL of glutaraldehyde type II (25% W in water) were added, and the mixture was stirred for 1 h at 120 rpm using an orbital stirrer (recommended for the first step S1).

At the end of step 1 (form S1: GA-activated enzymes), 4.4 mL of PTR-HCl pH 6.5 (30 mg/mL) was added and the mixture was subsequently stirred at 120 rpm for 30 min, to interconnect remaining aldehyde terminal groups present on the activated enzyme surface through a PTR cross-linking. Next, the pH was adjusted to 8.0 by adding (dropwise) 1 M NaOH aqueous solution and stirred in a rotator stirrer (recommended for the second step S2) for 10 min. Finally, 100 mL of 25 mM sodium phosphate buffer pH 7.0 was added and the suspension was brought back again to the rotator stirrer until the total copolymerization and formation of the catalyst (form S3, Figure 6). The pH and specific activity were determined at the end of each step during the preparation of the catalyst.

In the case of CAL-B*, the CLEA-LC obtained after the first adjustment of pH (≈8.0) was discarded, and then the pH was raised to 12 using 1 M NaOH and stirred (rotatory stirrer) until reaching the desired copolymerization level, which was determined by measuring the residual activity in the supernatant.

CLEA-LCs (form S3) were collected and compressed using centrifugation (8000 rpm, 15 min) to obtain the form S3*, which was then treated with 1 mg/mL NaBH_4_ fresh suspension to obtain the final form of our catalyst (form S4, Figure 6)

#### 4.2.6. SDS Page Electrophoresis

A 12% SDS page electrophoresis gel was prepared and utilised to show the difference between the free form of the enzyme (form S0) and the glutaraldehyde activated ones (form S1). Approximately 1.25 µg protein of both forms (free-form and activated-form) of S1 enzymes was utilised for the experiment. The results are shown in Figure 3.

#### 4.2.7. FT-Infrared of CRL CLEA-CL and Free CRL

FTIR-spectrum of dried CRL’s CLEA-LC and the free counterpart was performed using an Agilent Cary 630 FTIR.

#### 4.2.8. Scanning Electron Microscopy Analysis

Scanning electron microscopy (SEM) images of CRL CLEA-LC were taken using a HITACHI TM 1000 tabletop SEM. The CLEA-LC used the fully dried S4 form of the catalyst.

#### 4.2.9. Thermal Stability CRL CLEA-LC Compared with Free CRL

The thermal stability of both forms (free and CLEA-LC) of *Candida rugosa* lipase was studied at 25 °C, 40 °C, 50 °C and 60 °C during a period of 384 h (16 days). A total of 200 mg of CRL CLEA-LC was suspended in 10 mL of 25 mM sodium phosphate buffer pH 7.0 and dispersed using sonication for 3 min. Stability of CRL CLEA-LC was compared with the free form using the same amount of protein in both cases. The initial activity was taken as 100% at the beginning of the study as reference for the next residual activities.

#### 4.2.10. Preparation of Rac-Ibuprofen Ethyl Ester

Racemic ibuprofen (*rac*-ibuprofen) ethyl ester was prepared using a simple Fischer esterification. In brief, 0.5 mL of concentrated sulfuric acid was added to 40 mL of absolute ethanol. After 10 min of stirring, 2 g of *rac*-ibuprofen was added and the mixture was stirred for approximately 4 h at 40 °C. The reaction’s progress was followed using TLC (petroleum ether/ethyl acetate: 80/20: *v/v* as mobile phase and 254 nm UV lamp for substrate and product detection), until the total conversion to *rac*-ibuprofen ethyl ester. Then, the suspension was partially concentrated and 20 mL of *n*-hexane was added. The organic phase was washed three times with 20 mL of 0.5 M NaOH, then three times with Milli-Q distilled water and dried using anhydrous MgSO_4_ salt. The organic phase was concentrated and approximately 1.65 g (73% yield) of *rac*-ibuprofen ethyl ester was recovered. The purity of the product was confirmed using HPLC using analysis conditions described in Section 4.2.11.

#### 4.2.11. Kinetic Resolution of *rac*-Ibuprofen Ethyl Ester CLEA-LC and Free Enzymes: Comparative Study

A total of 30 mg of *rac*-ibuprofen ethyl ester was added to 10 mL of 10 mM tris-HCl pH 7.0. CLEA-LC of the six enzymes and their free counterparts were used with protein content as mentioned in Table 4. Mixtures were stirred at 250 rpm and 37 °C using an orbital shaker with controlled temperature for 24 h. Finally, reactions were stopped and pH was reduced to 5. Both residual ester and formed acid were extracted together using *n*-hexane. HPLC analysis was performed using Chiralcel ODH chiral column with flow rate of 1 mL/min, mobile phase *n*-hexane/isopropanol/TFA: 100/1/0.1 (*v/v/v*) and UV wavelength of 254 nm. All experiments were carried out in triplicate.

#### 4.2.12. Reuse of CRL CLEA-LC

CRL CLEA-LC was subjected to 5 cycles of 24 h using the same conditions mentioned in Section 4.2.11 in order to study the resistance of the CLEA-LC of this enzyme. After each cycle, the CLEA-LC was washed three times using a 10 mM tris-buffer pH 7.0 and centrifuged to be reused in the next cycle.

#### 4.2.13. Long-Term Storage of CLEA-LC

CRL and CAL-B CLEA-LCs (500 mg from S4 of each CLEA-LC) and their free counterparts with the same protein and water content were studied to test their storage stability at 4 °C during 3 months (90 days) by analysing their hydrolytic activities every 10 days intervals; thus, samples were withdrawn and tested in the hydrolysis *p*NPB (see Section 4.2.3).

## 5. Conclusions

We reported in this manuscript a simple and quick preparation of lipase-based CLEA-LCs (Cross-Linked Enzyme Aggregate-Like Copolymers) using one-pot, consecutive cross-linking steps using two types of homobifunctional reagents, glutaraldehyde (GA) and putrescine (PTR), via amine activation through pH alteration (pH jump) as a key step in the process. In fact, the employed methodology is simple and fast, as it only requires stirring with GA for 1 h, a subsequent addition of PTR and stirring during 30 min, pH adjustment to around 8.0 and 10 min stirring again with buffer pH 7.0.

To follow this methodology, CAL-B was used to set the best experimental conditions; subsequently, other commercial lipases were employed to prepare the corresponding CLEA-LCs. The optimised CLEA-LC preparation applied to various lipases mirrored by CAL-B. The described technique minimally affected catalytic properties, as evidenced by comparing the specific activities of the free form and the CLEA-LCs. Good activity retention was observed for CAL-B, CAL-B*, TLL, LU, and RML, with slight decreases in CRL (76%) and CAL-A (77%). The highest recovered activity (R.A, %) was observed in CAL-B* and RML (113.14% and 106%, respectively), while other enzymes showed good to high recovered activities from 64% to 76%.

The so-prepared CLEA-LCs were tested to check their capability to catalyse the kinetic resolution of racemic ibuprofen ethyl ester. Compared to the native enzymes, CLEA-LCs led to substantial enhancements in enantioselectivity and conversion for most cases, except CAL-B. CAL-B* exhibited a notable increase in conversion (25% to 45%) and enantiomeric excess (from 15 up to 48%), while LU and CAL-A also showed improved conversion and enantioselectivity upon immobilization. CRL-based CLEA-LC demonstrated exceptional catalytic performance, maintaining high conversion and enantioselectivity, surpassing the reported results for other immobilised CRL derivatives. Additionally, this CRL-based CLEA-LC exhibited significant thermal stability compared to the native enzyme, indicating a more stable enzyme molecule due to covalent multipoint attachment. Finally, CAL-B and CRL CLEA showed only slight decreases (around 8%) in specific activity after three months.

To sum up, the simplicity and ease of the experimental procedure required to prepare the CLEA-LCs for obtaining effective and stable lipase-based biocatalysts has been proven. Remarkable enantioselectivity and conversion enhancements were achieved, especially with CRL, underscoring CLEA-LCs’ potential as effective lipase-based biocatalysts. The method’s simplicity and excellent performance with CRL CLEA-LC for enantioselective hydrolysis of racemic ibuprofen ethyl ester highlight its utility. Thus, we envision that this methodology could be extended to other enzymes and other substrates, as it is very straightforward to carry out. Therefore, many potential applications could be fulfilled by testing the applicability of CLEA-LCs for the kinetic resolution of other racemic compounds, or even testing their behaviour in dynamic kinetic resolutions (DKRs). Nevertheless, for scaling this methodology up to higher levels, some other experiments (e.g., mechanical stability, performance in continuous mode, etc.) must be conducted to prove their true potential.

## Figures and Tables

**Figure 1 ijms-24-13664-f001:**
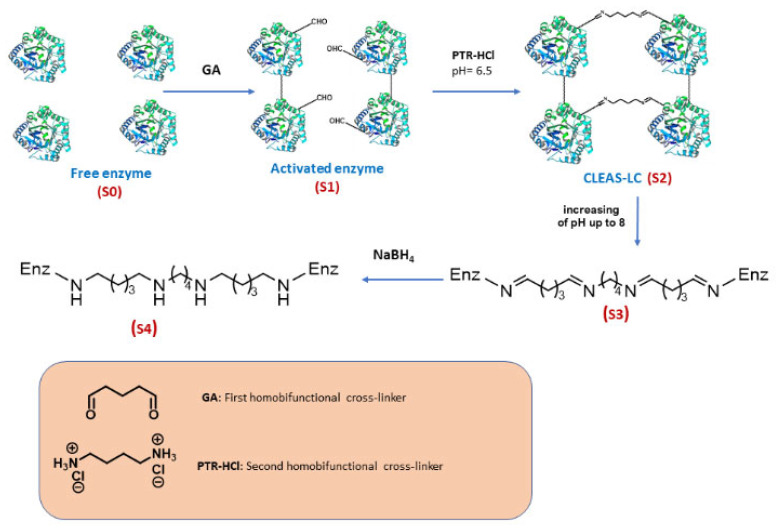
Schematic illustration of different steps of CLEA-Like Copolymer preparation. S0–S4 different states (or forms) of catalyst: S0: Free enzyme suspension, S1: GA-activated enzyme, S2: copolymer as liquid copolymer before pH Jump, S3: formation of solid copolymer induced by pH rising (amine activation), S4: NaBH_4_ treated CLEA-LC.

**Figure 2 ijms-24-13664-f002:**
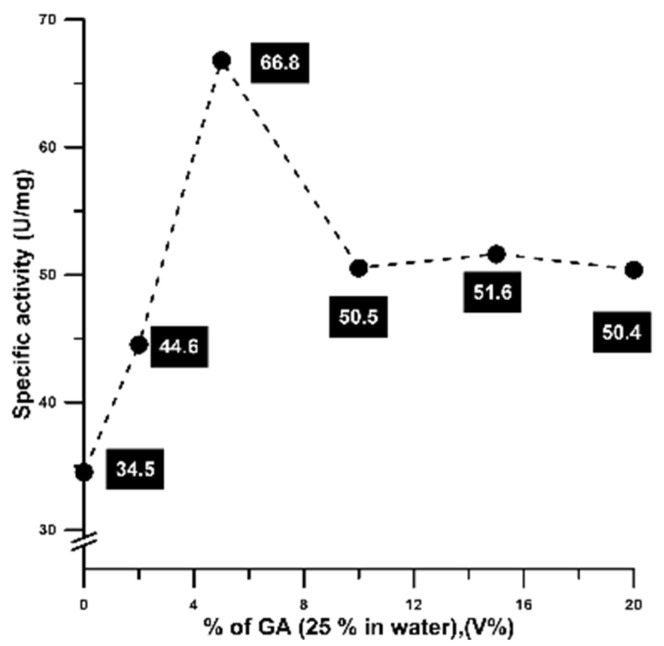
Effect of glutaraldehyde percentage on the resulting GA-Activated Lipase B from *Candida antarctica* (State S1 of the catalyst). Experimental error was approximately 5%.

**Figure 3 ijms-24-13664-f003:**
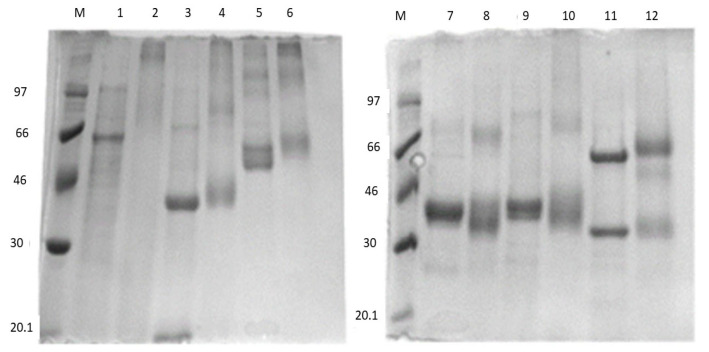
12% SDS-page electrophoresis gels of free (S0) and GA-activated enzymes (S1): M: molecular weight marker; 1: free CRL; 2: activated CRL; 3: free CAL-B; 4: activated CAL-B; 5: free CAL-A; 6: activated CAL-A; 7: free TLL; 8: activated TLL; 9: free LU; 10: activated LU; 11: free RML; 12: activated RML.

**Figure 4 ijms-24-13664-f004:**
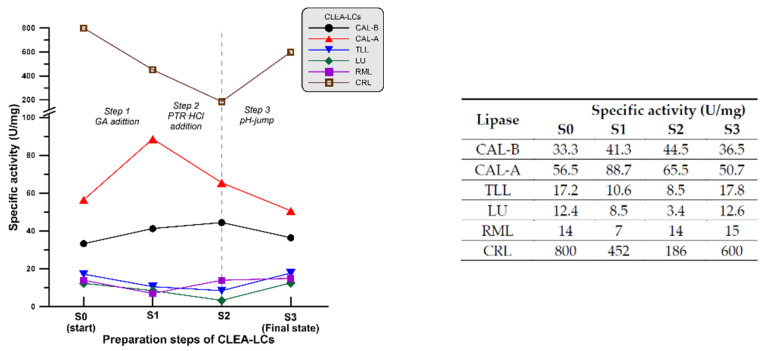
Specific activity during each step in the preparation of CLEA-LCs preparation. S0–S3 are the different states of the catalyst: S0: free enzyme, S1: GA-activated enzyme, S2: GA/PTR/Enzyme initial liquid copolymer, S3: final solid copolymer at the end of preparation and before centrifugation. Experimental error was approximately 5%.

**Figure 5 ijms-24-13664-f005:**
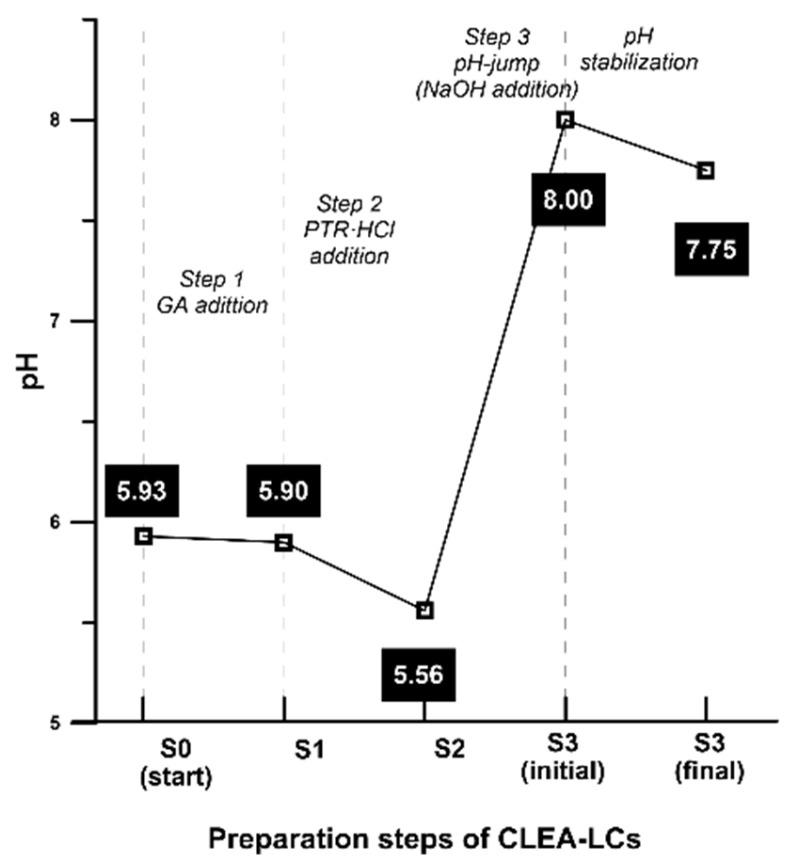
pH monitoring in the preparation of CRL CLEA-LC.

**Figure 6 ijms-24-13664-f006:**
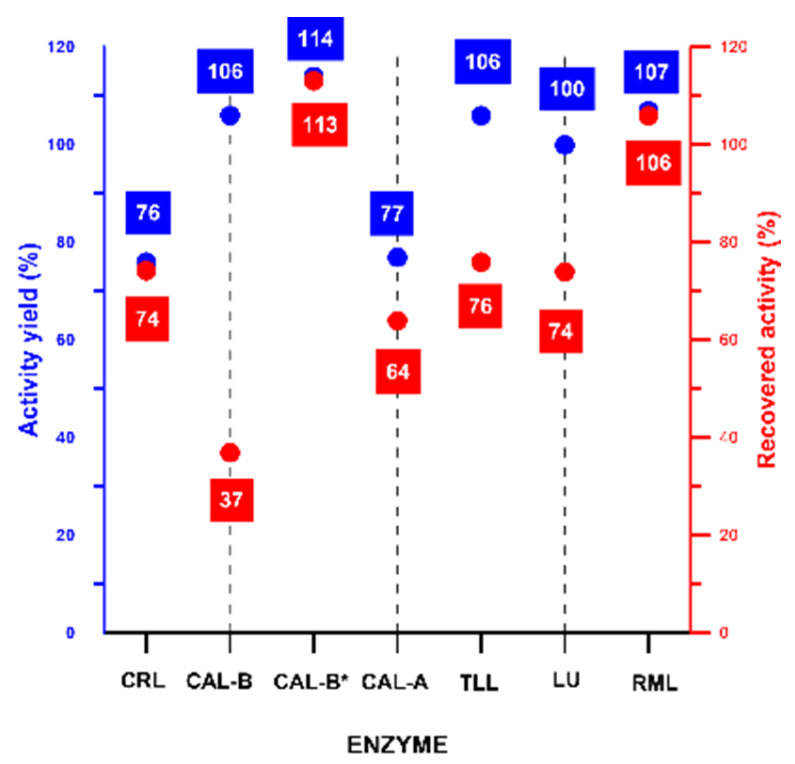
Activity yield (A.Y) and recovered activity (R.A) of each enzyme mentioned in Table 3. AY. (%): same formula as Table 3, R. A. (%) = [(total units in the CLEA-LC)/total initial units used in the preparation of CLEA-LC (of 50 mg of free enzyme)] × 100. Error was approximately 5%.

**Figure 7 ijms-24-13664-f007:**
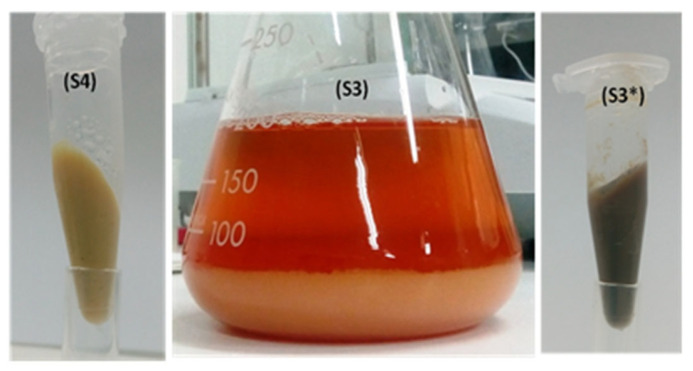
Final forms of CLEA-LC of lipase from *Candida rugosa*. S3 (third state of the catalyst) final form or the end of step 3; S3*: recovered CLEA-LC through centrifugation (8000 rpm, 15 min); S4: recovered CLEA-LC after treatment with freshly prepared 1 mg/mL NaBH_4_ in ddH_2_O (5 mL/1 g CLEA-LC) for 30 min with orbital steering followed by washing and centrifugation.

**Figure 8 ijms-24-13664-f008:**
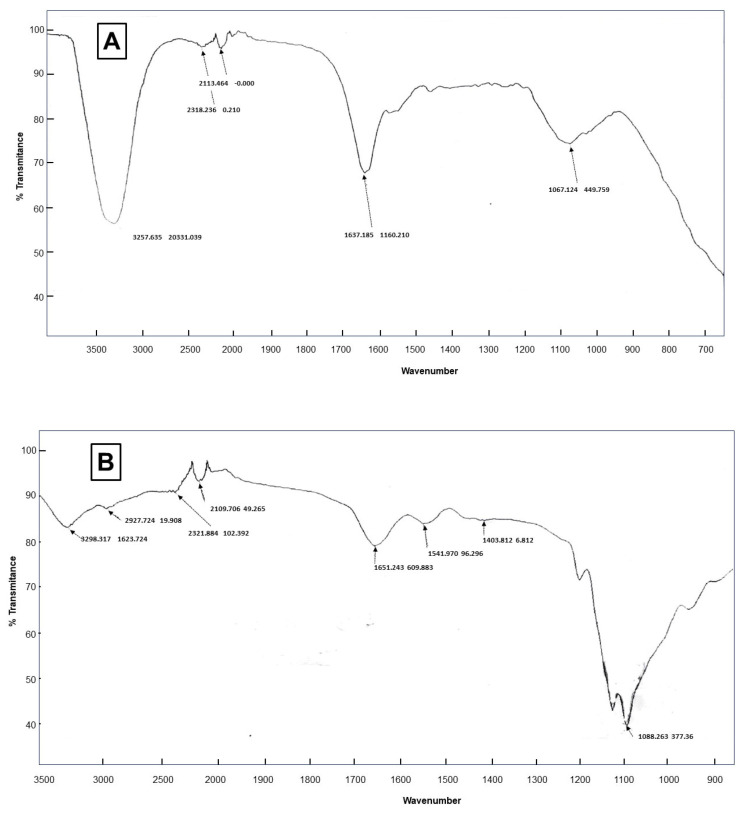
(**A**) FTIR Spectra of CLEA-LC from CRL (State S3*); (**A**,**B**) FTIR Spectra of native CRL.

**Figure 9 ijms-24-13664-f009:**
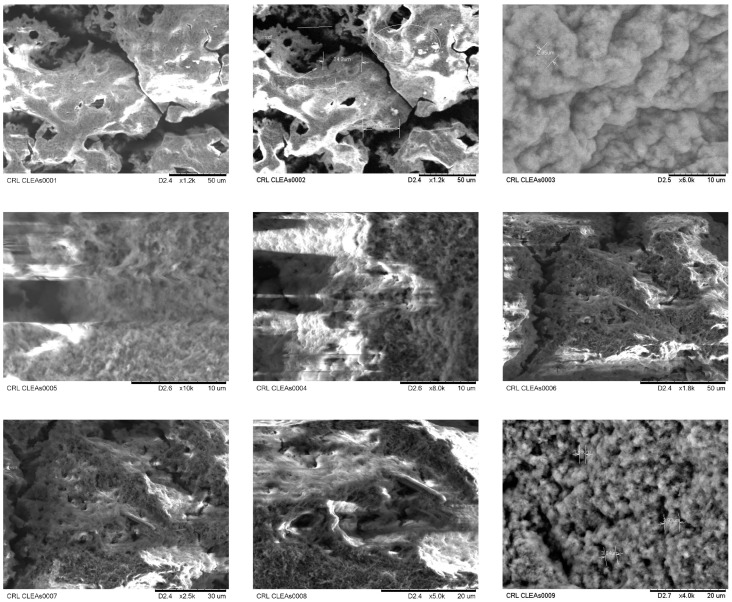
Scanning electron microscopy (SEM) images of CRL CLEA-LC using the dry S4 form of the catalyst.

**Figure 10 ijms-24-13664-f010:**
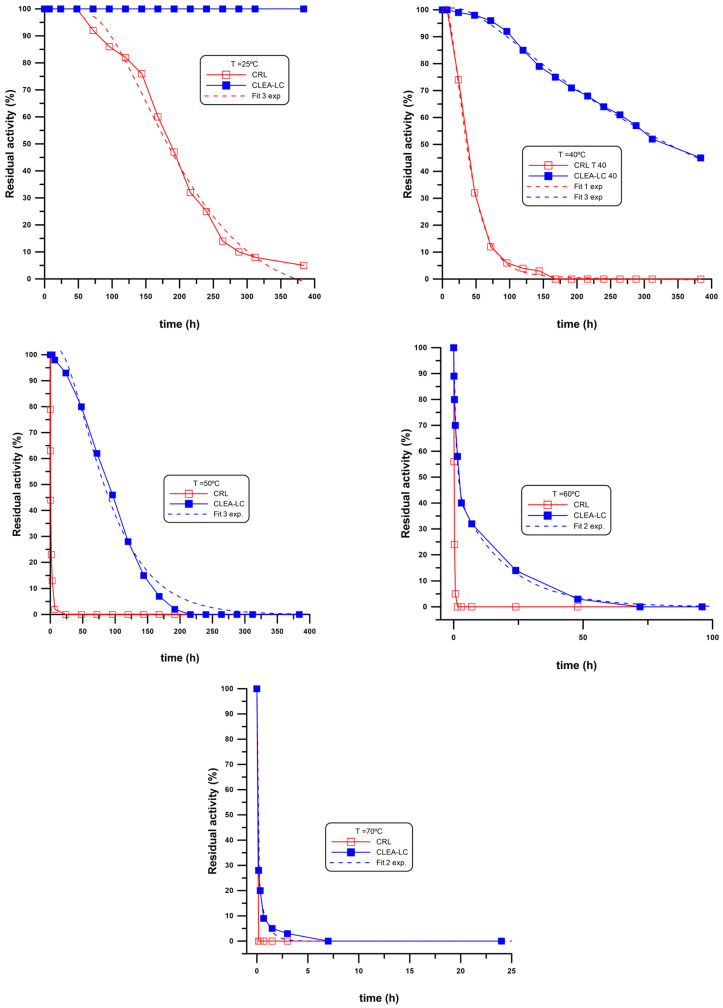
Thermal stability of CRL CLEA-LC compared with free CRL at 25 °C, 40 °C, 50 °C, 60 °C and 70 °C. A total of 200 mg of CRL CLEA-LC and the same amount of protein from free CRL in 10 mL of 25 mM sodium phosphate buffer pH 7.0. Residual activity (%) =[(At/A0) × 100].

**Figure 11 ijms-24-13664-f011:**
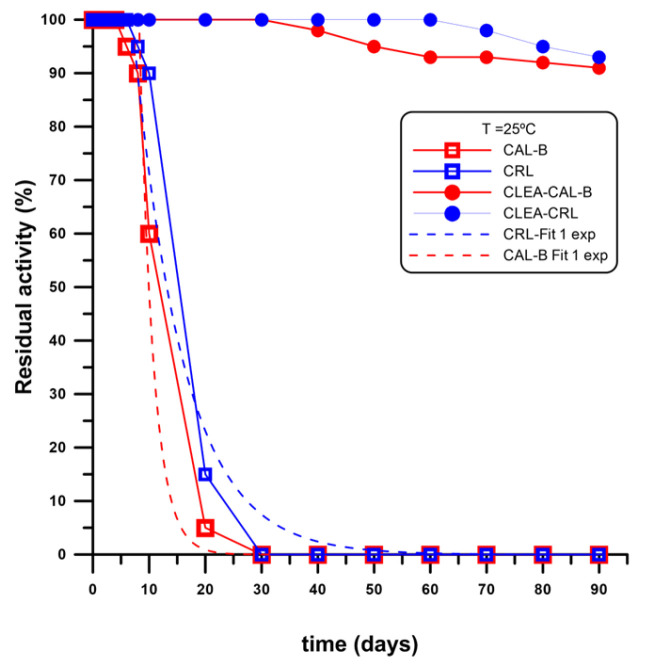
Storage stability CRL and CAL-B CLEA-LC compared with the corresponding counterpart suspended in Milli-Q ddH_2_O the same content water of the corresponding CLEA-LC and same amount of protein (enzyme) at 4 °C. Experiment was performed during a period of 3 months (90 days).

**Figure 12 ijms-24-13664-f012:**
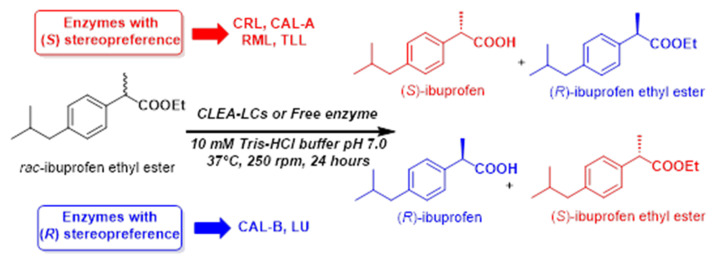
Kinetic resolution of *rac*-ibuprofen ethyl ester using various preparations of CLEA-LC.

**Figure 13 ijms-24-13664-f013:**
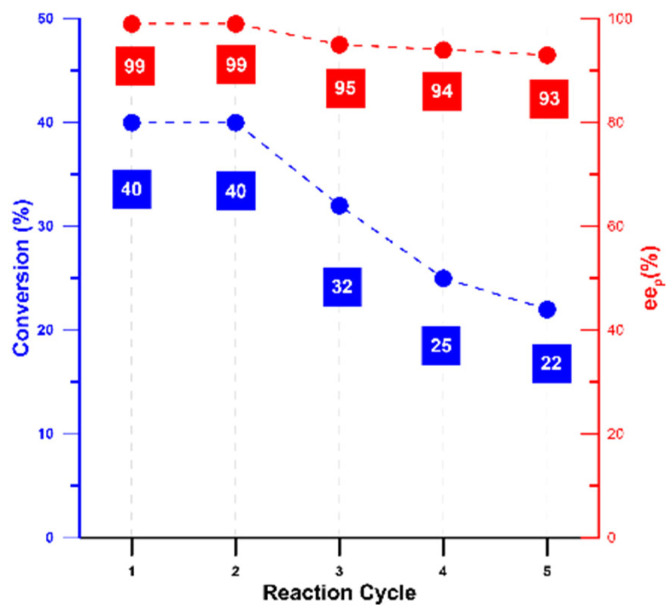
Reuse of CLEA-LC of *Candida rugosa* lipase: same condition as mentioned in Table 4.

**Table 1 ijms-24-13664-t001:** Preliminary preparation of CLEA-LCs using 5 mg of free CAL-B in 10 mL of 25 mM sodium phosphate buffer pH 7.0. Immobilization yield (I.Y, (%)), specific activity (S.A, U/mg) and activity yield (A.Y, %) are described in Section 4.2.4.

Group	Case	Molar Equivalents. CAL-B.: GA: PTR.	V _GA._/V% (µL)	V _PTR._ (µL)	I.Y. (%)	S.A. _CLEAs_ (U/mg)	A.Y. (%)
G1	01	1: 250: 250	15/0.15	110	10	35	100
G2	02	1: 500: 250	30/0.30	110	13	35	100
	03	1: 500: 500	30/0.30	220	15	34	96
	04	1: 500: 1000	30/0.30	440	18	34	97
G3	05	1: 1000: 250	60/0.60	110	15	35	100
	06	1: 1000: 500	60/0.60	220	19	35	100
	07	1: 1000: 1000	60/0.60	440	25	36	103
G4	08	1: 2000: 250	120/1.18	110	20	34	97
	09	1: 2000: 500	120/1.18	220	23	35	100
	** *10* **	** *1: 2000: 1000* **	** *120/1.18* **	** *440* **	**35**	**37**	**106**
	11	1: 2000: 2000	120/1.18	880	29	30	86

**Table 2 ijms-24-13664-t002:** pH jump S2–S3 resulted from dropwise adjustment with 1 M NaOH, measured using a calibrated pH meter.

Enzyme	pH Jump (S2–S3)
CRL	5.46–8.05
CAL-B	5.48–8.19
CAL-B*	8.19–12.0
CAL-A	5.81–8.02
LU	5.90–8.08
TLL	5.79–8.15
RML	5.98–8.08

**Table 3 ijms-24-13664-t003:** Preparation of CLEA-LCs in State S3* (final solid copolymer recovered after centrifugation at 8000 rpm, 15 min).

Lipases	Enzyme (mg)	I.Y (%)	W_CLEA-LC_ S3* (g)	S.A. Free Enzyme (U/mg)	S.A. CLEA-LC S3* (U/mg)	CLEA-LC Activity S3* (U/g)	%H_2_O S3*
CRL	50	99	8.5	800	600	3494	93.9
CAL-B	50	35	1.36	35	37	476	93.8
CAL-B*	50	>99	6.5	35	>40	307	93.7
CAL-A	50	83	1.73	65	50	1199	93.8
TLL	50	72	1.4	17	18	468	94.2
LU	50	74	1.47	12	12	302	93.1
RML	50	99	1	14	15	745	93.7

Preparation of CLEA-LC using 50 mg of each free enzyme suspended in 100 mL of 25 mM sodium phosphate buffer pH 7.0. (0.5 mg/mL), I.Y, S.A, as reported in Section 4.2.4. WCLEA-LC S3*: compressed wet weight of obtained CLEA-LC; S.A. CLEA-LC S3*: specific activity of enzyme present in the CLEA-LC in state S3*), CLEA* (U/g): activity units per gram of compressed wet CLEA-LC. %H_2_O: entrapped water in the compressed wet CLEA-LC).

**Table 4 ijms-24-13664-t004:** Kinetic resolution of *rac*-ibuprofen ethyl ester using the free and corresponding CLEA-LC counterpart with the same enzyme content from each preparation.

Enzyme	Free Form			CLEA-LC Form		
	C (%)	ee_p_ (%), SP.	E	C (%)	ee_p_ (%), sp.	E
CRL	49	97, (S)	>200	40	99, (S)	>200
CAL-B	76	27, (R)	----	64	30, (R)	----
CAL-B*	25	15, (R)	----	45	48, (R)	8.47
CAL-A	23	22, (S)	3	46	66, (R)	14.30
RML	81	22, (S)	----	47	44, (S)	9.24
LU	8	06, (R)	2	33	47, (R)	5.56
TLL	74	75, (S)	----	3.4	>99	2.24

Conditions: CRL (0.115 mg), CAL-B (0.115 mg), CAL-B* (0.045 mg), CAL-A (2.4 mg), TLL (5.14 mg), LU (1.26 mg), RML (2.5 mg), 30 mg of substrate (*rac*-ibuprofen ethyl ester), 10 mL of 10 mM Tris-HCl 7.0, 250 rpm, 37 °C, 24 h. C (%): overall conversion, ee_p_: enantiomeric excess of the product (*R* or *S* ibuprofen), SP.: stereopreference of the catalyst; E: enantioselectivity (enantiomeric ratio, ratio between the specificity constants (kcat/KM) for both enantiomers), calculated as E = ln[(1 − C) × (1 + ee_p_)]/ln[(1 − C × (1 − ee_p_)], ee_p_ and C as decimals (≤1) [65,66]).

**Table 5 ijms-24-13664-t005:** Deactivation parameters calculated from plots in Figure 9, according to a sum of exponential functions (Y = A_n_·(exp(−k_n_)·X).

	T (°C)	t_1/2_ (h)	A_1_	A_2_	A_3_	k_1_	k_2_	k_3_	R^2^
CRL	25	192	−1.11 × 10^3^	8.76 × 10^1^	1.13 × 10^3^	1.33 × 10^−2^	1.21 × 10^−2^	1.19 × 10^−2^	0.9836
	40	35	1.05 × 10^2^	---	---	2.29 × 10^−2^	---	---	0.981
	50	0.67	9.56 × 10^1^	---	---	1.06	---	---	0.9891
	60	0.5	1.01 × 10^2^	---	---	4.01			0.9977
	70 ^a^	---	---	---	---	---	---	---	---
CLEA-LC	25 ^b^	---	---	---	---	---	---	---	---
	40	312	−1.66 × 10^1^	−1.23 × 10^1^	1.29 × 10^2^	2.51 × 10^−2^	2.51 × 10^−3^	2.51 × 10^−3^	0.9976
	50	96	−3.12 × 10^2^	1.39 × 10^1^	3.97 × 10^2^	2.86 × 10^−2^	1.98 × 10^−2^	1.99 × 10^−2^	0.9961
	60	2	4.65 × 10^1^	5.15 × 10^1^	---	5.30 × 10^−2^	1.02	---	0.9986
	70	<0.5	2.80 × 10^1^	7.20 × 10^1^	---	1.33	1.52 × 10^1^		0.9984

^a^ Deactivation too fast to be measured. ^b^ Deactivation not observed during the experiment.

## Data Availability

Not applicable.

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
