# Peer review of "Easy and Versatile Technique for the Preparation of Stable and Active Lipase-Based CLEA-like Copolymers by Using Two Homofunctional Cross-Linking Agents: Application to the Preparation of Enantiopure Ibuprofen"

_ijms, 2023, doi:10.3390/ijms241713664_

Round 1

Reviewer 1 Report

This paper deals with the enzyme aggregates by cross-linking of lipase.  The paper has following problems:

     Cross-linking formed by glutaraldehyde (GA) is quite commonly technique, so that the authors should describe the novelty point of the proposed method.  Aggregation formation by GA is a quite old method (it used about 20 years ago.).  The authors should show the references and describe the development point of the presented method.

     The authors should describe why GA was used?

     Merit of the proposed method is not clear.

     In Fig. 1, the authors show enzyme would bind via PTR.  However, direct binding between enzymes is formed in this case.  If the authors want to say it, the evidence should be shown.

     In Title, the authors wrote word “improved properties”.  What properties are improved?

     As shown in Fig. 2, binding with GA loses the enzyme activity so much.  How the authors think about?

From above reasons, the paper should be rejected.

There are some sentences that the readers cannot understand it.  The authors should check the whole manuscript.

Reviewer 2 Report

"Easy and versatile technique for the preparation of lipase CLEA-Like Copolymers with improved properties" by Oussama Khiari, Nassima Bouzemi, José María Sánchez-Montero and Andrés R. Alcántara is an original research article. Reactions of enzymatic hydrolysis are very important branch of ambient conditions catalysed reactions. Among of these some examples can be given, such us hydratase amidase, nitrilase, diolase and invertase. The authors prove the immobilisation of the enzyme isolated from Candida Rugosa, the lipase. They use onvincing array of techniques to evidence immobilisation and demonstate enantioselectivity of hydrolysis of racemate of ibuprofen ethyl esters. The presented research is a firm step further into one pot immbobilisation of important enzymes for enantioselective catalysis. Therefore the paper is recommended for publication in IJMS, after placing the Conclusions section into the manuscript, which is currently lacking.

Reviewer 3 Report

Reviewer report on manuscript ‘Easy and versatile technique for the preparation of lipase CLEA‐Like Copolymers with improved properties’

In this work authors Enzyme‐aggregates Like Copolymers (CLEA‐LC) were formed by cross‐linked using two types of homo-bi-functional cross‐linkers (e.g. glutaraldehyde) mediated by amine activation through pH‐alteration (pH‐jump) as a key step in the process. Six enzymes were utilized as models in order to investigate the effectiveness of the technique mainly in term of immobilization yields, hydrolytic activities, thermal stability and application. A good retention of catalytic properties was found with almost all of cases beside an important thermal and storage stabilities improvement.

Manuscript is well supported by data, it is well written and could be interesting for researchers who are working in area of biosensorics and biotechnology. Therefore, manuscript can be published after below indicated minor corrections and improvements:

Abstract needs some improvements in order more carefully address finding reported in this manuscript.

Introduction and discussion of results could be advanced taking into account other references on immobilization of enzymes by glutaraldehyde ( Amperometric biosensor for the determination of creatine. Analytical and Bioanalytical Chemistry 2007, 387, 1899–1906. // Glucose biosensor based on graphite electrodes modified by glucose oxidase and colloidal gold nanoparticles. Microchimica Acta 2010, 168, 221229.), which could be overviewed ad discussed in Introduction, Discussion and in other parts of the manuscript.

Data presented in Figures 4, 5 and 6 better fits to the Table instead of recently used graphical-plots. In graphical-plots better present data, which have some interdependencies, that can be described by some particular function. Deviation of these results could be presented in tables.

Conclusions and future trends should be addressed at the end of the discussion part of the manuscript.

N/A

Reviewer 4 Report

In the manuscript entitled “Easy and versatile technique for the preparation of lipase CLEA-Like Copolymers with improved properties”, enzyme-aggregates Like Copolymers (CLEA-LC) by one-pot, consecutive cross-linking steps using two types of homobifunctional cross-linkers mediated by amine activation through pH-alteration (pH-jump) as a key step in the process. Six enzymes were utilized as models in order to investigate the effectiveness of the technique mainly in term of immobilization yields, hydrolytic activities, thermal stability and application. A good retention of catalytic properties was found with almost all of cases beside an important thermal and storage stabilities improvement.

 In general, the study is well-conducted and provided some important findings. However, some revisions are required as shown below;

 - The abstract should highlight the most important findings of this study.

 - The introduction is not discussing the review of literature and hypothesis behind the topic of this study. Please cite and discuss the recent studies and literature. Additionally, the objective of the study should be mentioned in more details at the end of the introduction section.

 - Material and Methods:

- Number of sample or replicates should be mentioned for each experiment in the methods!

 - Results are clear and well-represented.

 - The discussion should be interpreted with the results as well as discussed in relation to the present literature.

 -The conclusion section should be re-written to highlight the significant findings and recommendations of this study as well as to mentioned the future perspectives.

 - The references section should be updated as per my above-mentioned suggestion

Moderate editing of English language required

Reviewer 5 Report

The manuscript submitted by Khiari et al., titled: "Easy and versatile technique for the preparation of lipase CLEA-Like Copolymers with improved properties." is an interesting work aiming to develop an alternative method for the generation of copolymers with improved properties.

The paper is well written and flows nicely for the reader. The reviewer would like to offer a few points below for the authors' consideration:

1. Consider adding a short paragraph on potential applications for the product generated and how this new method may improve a potential supply chain conceptually if it were to upgrade to industrial scale.

2. Consider outlining more clearly and specifically the rationale for the conditions and concentrations used for the development of the method.

3. Consider commenting more extensively on potential limitations and challenges with the proposed approach.

Nice work overall.

English language is OK overall. Proofreading is suggested.

Round 2

Reviewer 1 Report

This paper deals with the CLEA by some lipases.  The paper would contain new results, however, it is not clear that what function (and/or property) is improved compared to the other immobilization methods.  The authors should discuss the development, along with the purpose of the immobilization method.

Some sentences in the manuscript would be difficult to understand.  The authors should check the whole manuscript carefully.
